# *Trichoderma viride* Colonizes the Roots of *Brassica napus* L., Alters the Expression of Stress-Responsive Genes, and Increases the Yield of Canola under Field Conditions during Drought

**DOI:** 10.3390/ijms242015349

**Published:** 2023-10-19

**Authors:** Zuzanna Garstecka, Marcel Antoszewski, Agnieszka Mierek-Adamska, Daniel Krauklis, Katarzyna Niedojadło, Beata Kaliska, Katarzyna Hrynkiewicz, Grażyna B. Dąbrowska

**Affiliations:** 1Department of Genetics, Faculty of Biological and Veterinary Sciences, Nicolaus Copernicus University in Toruń, Lwowska 1, 87-100 Toruń, Poland; z.znajewska@gmail.com (Z.G.); mant@doktorant.umk.pl (M.A.); mierek_adamska@umk.pl (A.M.-A.); 2Research Centre for Cultivar Testing in Słupia Wielka, Chrząstowo 8, 89-100 Nakło nad Notecią, Poland; 3Department of Cellular and Molecular Biology, Faculty of Biological and Veterinary Sciences, Nicolaus Copernicus University in Toruń, Lwowska 1, 87-100 Toruń, Poland; karask@umk.pl; 4Department of Microbiology, Faculty of Biological and Veterinary Sciences, Nicolaus Copernicus University in Toruń, Lwowska 1, 87-100 Toruń, Poland

**Keywords:** *Trichoderma*, canola, field trial, bioinoculants, metallothionein, stringent response, *RSH*, PGPM, promoter

## Abstract

In this work, we present the results of the inoculation of canola seeds (*Brassica napus* L.) with *Trichoderma viride* strains that promote the growth of plants. Seven morphologically different strains of *T. viride* (TvI-VII) were shown to be capable of synthesizing auxins and exhibited cellulolytic and pectinolytic activities. To gain a deeper insight into the molecular mechanisms underlying canola–*T. viride* interactions, we analyzed the canola stress genes metallothioneins (*BnMT1-3*) and stringent response genes (*BnRSH1-3* and *BnCRSH*). We demonstrated the presence of *cis*-regulatory elements responsive to fungal elicitors in the promoter regions of *B. napus MT* and *RSH* genes and observed changes in the levels of the transcripts of the above-mentioned genes in response to root colonization by the tested fungal strains. Of the seven tested strains, under laboratory conditions, *T. viride* VII stimulated the formation of roots and the growth of canola seedlings to the greatest extent. An experiment conducted under field conditions during drought showed that the inoculation of canola seeds with a suspension of *T. viride* VII spores increased yield by 16.7%. There was also a positive effect of the fungus on the height and branching of the plants, the number of siliques, and the mass of a thousand seeds. We suggest that the *T. viride* strain TvVII can be used in modern sustainable agriculture as a bioinoculant and seed coating to protect *B. napus* from drought.

## 1. Introduction

*Brassica napus* L. (rapeseed, canola, or oilseed rape) is one of the most important crops, being the second most widely grown oilseed crop in the world. Rapeseed oil has a well-balanced amino acid profile and desirable features of technological and functional utility for industry [1]. Rapeseed oil production is continuously growing due to its wide range of applications; i.e., it is used for human and animal nutrition and in the manufacture of biofuels, paints, polyurethane, and other products [2,3]. In Europe, over 24 million tons of rapeseed is produced annually, accounting for 32.9% of global rapeseed production. Germany, France, and Poland are the three biggest European producers of rapeseed [4]. 

Progressing climate change leading to drought events that are longer and more frequent is negatively affecting the growth and yield of crops [5,6]. Drought inhibits metabolic processes and thereby reduces biomass accumulation and disrupts seed formation [7]. It is now widely known that the plant microbiome is crucial for plant stress resistance [8]. Among the great variety of microorganisms that interact with plants, there are mycorrhizal fungi, which are able to colonize the roots of around 85% of vascular plants [9]. Mycorrhizal fungi can increase plant stress resilience [10,11] under various adverse conditions, including drought [12,13]. The lack of interaction of some plant species with mycorrhizal fungi is probably caused by the absence of anatomical adaptations in plant roots or secreting antifungal compounds in root exudates (e.g., isothiocyanates, defensins, and glucosinolates), which can inhibit the development of fungal spores and mycelium [14]. Among the plants that do not interact with mycorrhizal fungi are species of the families *Chenopodiaceae*, *Amaranthaceae*, and *Brassicaceae*, including canola. Filamentous plant-growth-promoting fungi (PGPF) can be used to support the cultivation of non-mycorrhizal plants. Fungal inoculates are becoming more and more popular because they are an environmentally friendly alternative to agrochemicals. Agrochemicals negatively affect the natural environment and human health [15,16,17,18,19,20,21] and raise the costs of plant production in comparison to biological agents [22]. 

Fungi belonging to the genus *Trichoderma* promote the growth and development of monocotyledonous and dicotyledonous plants, e.g., increasing yields [23,24], increasing micro-/macro-element content [25], supporting seed germination and vigor [26,27], and improving the host plant’s functioning under both abiotic and biotic stresses [28,29,30,31]. *Trichoderma* can support plant growth via the production of phytohormones [32] and siderophores [28], the solubilization and mineralization of micro- and macro-elements [33], and the degradation of the precursor of ethylene—ACC (1-aminocyclopropane-1-carboxylate) [34]. *Trichoderma* effectively limits the development of plant pathogens by competing within the niche occupied by phytopathogens and by secreting peptaibols, antibiotics, and other metabolites, including volatile organic compounds (VOCs) [23,35]. *Trichoderma* secretes elicitors and changes phytohormone levels in the host plant, which trigger induced systemic resistance (ISR) and/or systemic acquired resistance (SAR) [36,37] and can affect the plant’s proteome [38] and secretome [39]. 

Various mechanisms are activated to enable a plant to adapt to stress conditions. Metallothioneins (MTs), low-molecular-weight proteins rich in cysteine residues (Cys), are examples of proteins related to different types of environmental stress. Promoters of *MT* genes contain several stress-response motifs, including elements responsive to microbial elicitors and drought, for example, in rice (*Oryza sativa* L.), *Arabidopsis thaliana* (L.) Heynh. [40], maize (*Zea mays* L.) [41], tobacco (*Nicotiana tabacum* L.), and oat (*Avena sativa* L.) [42,43]. Plant MTs (pMTs) are divided into four types, depending on the number and arrangement of cysteine residues: pMT1 (12 Cys), pMT2 (14 Cys), pMT3 (10 Cys), and pMT4 (17 Cys) [44]. MTs bind heavy metal ions through the -SH groups of cysteines and are involved in maintaining micronutrient homeostasis (e.g., zinc and copper) and the detoxification of toxic metals (e.g., cadmium and lead). In addition, thiol groups can be oxidized, and MTs are thus involved in the scavenging of reactive oxygen species (ROS). The expression of p*MT*s is dependent on the phase of plant development, is tissue-specific, and varies depending on the type of pMT. Plant MTs are hypothesized to be involved in processes such as the regulation of gene expression, metalloenzyme activity, apoptosis, DNA repair, cell growth and division, root development, fruit maturation, and cell wall suberization [45,46,47]. There are some clear indicators that pMTs are related to plant tolerance under water-limited conditions. For example, the expressions of p*MT*s were up-regulated by drought in rice [48], cotton (*Gossypium hirsutum* L.) [49], *A. thaliana* [50] oat [43], and guar (*Cyamopsis tetragonoloba* (L.) Taub.) [51]. Moreover, it was shown that the *MT* gene is up-regulated during dehydration and down-regulated during rehydration in the resurrection plant *Xerophyta humilis* (Baker) T. Durand & Schinz [52]. In the *B. napus* genome, 16 *MT* genes were identified, i.e., 1 *pMT1* gene, 10 *pMT2* genes, 3 *pMT3* genes, and 2 *pMT4* genes [44]. The other mechanism responsible for the regulation of plant development and the adaptability of plants is the stringent response. RSH proteins (RelA/SpoT homolog) synthesize and/or hydrolyze atypical nucleotides—alarmones (guanosine penta-tetra- and triphosphates; (pp)pGpp). The plant stringent response can be triggered by wounding, infection, salinity, treatment with abscisic acid (ABA), changes in light conditions, drought stress, and oxidative stress [53,54]. Plant RSH proteins are divided into three subtypes: RSH1, RSH2/RSH3, and Ca^2+^-activated CRSH. Plant RSHs are bifunctional proteins containing both alarmone synthetase (SYNTH) and hydrolase (HD) domains. The SYNTH domain mediates the addition of ATP-derived pyrophosphate to the 3′ end of GTP/GDP. Through Mn^2+^-dependent hydrolysis, which is mediated by the HD domain, (p)pGpp is converted to GTP/GDP and PPi [55]. In *B. napus*, similarly to *Arabidopsis*, RSH1 acts only as an alarmone hydrolase because of the lack of crucial Gly for the SYNTH domain, whereas CRSH acts only as a synthase due to the lack of His and Asp for the HD domain. RSH2 and RSH3 in *B. napus* possess functional SYNTH and HD domains [56]. Other conserved domains present in plant RSHs are the EF-hand motif in CRSH that is responsible for binding Ca^2+^ ions [57] and the ACT and TGS domains allowing for the recognition and binding of ligands [58]. At the N-terminus of plant RSH proteins, there is also a chloroplast-targeting sequence [55]. The latest reports suggest that the stringent response is activated in chloroplasts and changes the expression profile of plastidial and nuclear genes to fine-tune plant metabolism under adverse environmental conditions [59,60]. Under physiological conditions, plant RSHs regulate processes such as translation and photosynthesis [61], as well as affecting the levels of metabolites [62], lipids [63], and phytohormones [64]. RSHs play key roles in plant development and growth, especially during seed production [65] and plant aging [61]. The rapeseed genome contains 12 *RSH* genes: 6 *RSH1* genes, 2 *RSH2* genes, 3 *RSH3* genes, and 1 *CRSH* gene. Moreover, an in silico analysis of the promoter regions of rapeseed *RSH* genes showed the presence of *cis*-regulatory elements involved in the response to biotic factors [56]. 

Preliminary studies confirmed that *T. viride* promotes the growth of *B. napus* seedlings under laboratory conditions [26]. This study aimed to investigate the potential role of MTs and RSHs in plant–microbe interactions via an analysis of the expressions of *B. napus* metallothionein (*BnMT1*–*3*) and stringent response (*BnRSH1*–*3* and *CRSH*) genes. In silico analyses of the promoter regions of *BnMTs* and *BnRSHs* revealed the presence of *cis*-regulatory elements related to the plant response to fungal elicitors and phytohormones involved in a plant’s immune response, which indicated the potential role of BnMTs and BnRSHs in plant–fungi interactions. Further, to go beyond laboratory conditions for analyzing plant–microbe interactions, we tested whether the presence of *T. viride* affects the growth and yield of rapeseed under field conditions. To select a fungal strain for testing under field conditions, we determined the ability to produce auxin and hydrolytic enzymes, as well as the ability to colonize canola roots. Based on the obtained data presented herein, strain TvVII was selected, and the positive effect of this strain on the growth and yield of *B. napus* under adverse environmental conditions was proved. 

## 2. Results

### 2.1. Characterization of T. viride Strains

#### 2.1.1. Morphology

The *T. viride* strains showed significant differences in mycelium morphology. The pigmentation of the mycelium ranged from white, through yellow-green, to dark green (Figure 1). Moreover, the concentric rings were visible for some strains (TvI, TvV, and TvVI), whereas for other strains, nodules on the surface of the hyphae-specific conidiophore agglomerations in the form of lumps (TvI and TvV–TvVII) were present (Table 1). Some strains were characterized by a specific coconut smell (TvII, TvIII, and TvV), probably due to the production of the volatile compound 6-n-pentyl-2H-pyran-2-one. Spherical conidia were observed in six strains, while elliptical conidia were present in strain TvII (Table 1).

#### 2.1.2. Capacity to Synthesize Indole-3-Acetic Acid

All of the analyzed *T. viride* strains showed the ability to synthesize indole-3-acetic acid (IAA) in media not supplemented and supplemented with L-tryptophan (Figure 2). The level of IAA was the highest for strain TvII in the media supplemented with L-tryptophan, whereas in the media without L-tryptophan, this strain produced 4 times less IAA. Strains TvI, TvIII, and TvVII produced the same amount of IAA in both media. Interestingly, TvIV synthesized more IAA in the media without the addition of L-tryptophan than in the media containing this amino acid, i.e., around 3 times more. TvVII produced the least amount of IAA in comparison with the other strains, almost 5 times less than TvI and TvII. Only three strains, i.e., TvII, TvV, and TvVI, synthesized more IAA in the media containing L-tryptophan.

#### 2.1.3. Hydrolytic Activity

All analyzed strains produced hydrolytic enzymes hydrolyzing carboxymethylcellulose and pectin (Figure 3). The cellulolytic activity was the highest for the TvIII, TvV, and TvVI strains, whereas strains TvI, TvIV, TvV, and TvVII showed the highest pectinolytic activity. TvVII showed the lowest cellulolytic activity among the tested strains, whereas the pectinolytic activity of this strain was one of the highest. Similar results were observed for the other strains; i.e., strains showing high cellulolytic activity had low pectinolytic activity, and vice versa. Only the TvII strain showed relatively low cellulolytic and pectinolytic activities (Figure 3).

### 2.2. The Growth of B. napus Seedlings Inoculated with T. viride

In the next experiment, it was examined how the inoculation of *B. napus* seeds with *T. viride* spores of seven strains affected the growth of seedlings (Table 2). The roots of six-day-old seedlings that sprouted from seeds inoculated with fungal spores were longer in all variants, with the differences being statistically significant for the *T. viride* isolates TvI, TvII, TvVI, and TvVII. The longest roots were noted in the seedlings inoculated with TvVII, i.e., almost 1.5-fold longer than in the control, followed by TvI and TvII. The hypocotyls of the seedlings that grew from the seeds inoculated with the spores of TvI, TvII, and TvV–VII were longer than in the control. The hypocotyls of the seedlings inoculated with TvII were the longest, i.e., 1.5-fold longer than in the control, followed by TvVII and TvV. Interestingly, the hypocotyls of the seedlings that grew from the seeds inoculated with TvIII and TvIV were significantly shorter than the hypocotyls of the non-inoculated seedlings; however, other parameters were the same as those in the control seedlings. Both the fresh biomass and dry biomass of the seedlings that grew from the seeds inoculated with TvVI and TvVII were higher than those of the control seedlings; i.e., the fresh biomass was 1.3- and 1.1-fold higher, respectively, and the dry biomass was around 1.2-fold higher (Table 2).

For further field experiments, out of seven *T. viride* strains analyzed under laboratory conditions, TvVII was selected because it increased the fresh biomass and dry biomass the most and induced the strongest stimulation of the shoot and root growth of the *B. napus* seedlings.

### 2.3. The Colonization of B. napus Roots by T. viride

To verify the ability of *T. viride* to colonize the roots of *B. napus* seedlings, microscopic observations of the growth of fungal spores on the roots were performed (Figure 4). For the experiment, *T. viride* strain VII was selected because of the ability of this strain to promote the growth of rapeseed, as shown previously [26] and confirmed in this study. The microscopic observations revealed that fungal spores germinated toward the root after 24 h (Figure 4B). Within 48 h after inoculation (Figure 4C,D), hyphae started forming a net around the main root, indicating that, in some places, the hyphae penetrated the apoplast of the endoderm. After 72 h, the hyphae through entwinement formed a net around the root hairs (Figure 4E,F).

### 2.4. Analysis of BnMT and BnRSH Genes

#### 2.4.1. In Silico Analysis of Gene Promoters

An in silico analysis of the promoter regions of *BnMTs* and *BnRSHs* using PlantCARE software (https://bioinformatics.psb.ugent.be/webtools/plantcare/html/, accessed on 16 October 2023) [66] showed the presence of *cis*-regulatory elements involved in the response to elicitors and metabolites of fungal origin (Table 3). Those regulatory elements were the most abundant in the promoters of *BnMT2* and *BnCRSH*. The presence of elements for the response to phytohormones, including jasmonic acid (JA) and salicylic acid (SA), was also confirmed. In *BnMT1-3*, JA- and SA-responsive elements were present, whereas, in *BnRSH2*, *BnRSH3,* and *CRSH,* elements for the response to JA were noted.

#### 2.4.2. *B. napus* Metallothionein Gene Expression

The expression of *BnMT1-3* genes was evaluated in 6-day-old seedlings that grew from seeds inoculated with the spores of *T. viride* strain TvI-TvVII and compared with the expression of *BnMT1-3* genes in seedlings that grew from non-inoculated seeds (Figure 5). The relative level of *BnMT1* mRNA was not significantly affected by *T. viride* inoculation in cotyledons, whereas, in roots, the level of *BnMT1* transcripts was significantly increased by all tested *T. viride* strains. The highest expression of *BnMT1* was observed in the roots of the seedlings that grew from the seeds inoculated with TvV, i.e., 2.25-fold higher than in the control seedlings (Figure 5A). The mRNA level of *BnMT2* was significantly higher in the cotyledons of the seedlings that grew from the seeds inoculated with TvIII–VI spores, i.e., approximately 1.6-fold higher than in the control seedlings. The *BnMT2* mRNA level was significantly higher in the roots of the seedlings that grew from the seeds inoculated with TvV–VII spores, i.e., approximately 1.4-fold higher than in the roots of the control seedlings (Figure 5B). The expression of *BnMT3* in cotyledons increased in the seedlings that grew from the seeds inoculated with all tested fungal isolates, except for TvVII. The increase was the greatest for the TvIV and TvV strains, i.e., a 2.2-fold higher expression than in the control seedlings. In the roots, the expression of *BnMT3* was increased only in the seedlings that grew from the seeds inoculated with TvV; i.e., the expression was almost 2 times higher than in the control seedlings (Figure 5C).

#### 2.4.3. *B. napus RSH* Genes Expression

The inoculation of seeds with the spores of *T. viride* strains I–VII did not significantly increase the relative mRNA level of the stringent response genes in the cotyledons and roots of the 6-day-old rape seedlings in comparison to the control seedlings (Figure 6). In fact, only the expression of *BnRSH3* increased approximately 1.7-fold in the roots of the seedlings that grew from the inoculated seeds in comparison to the control (Figure 6C). Moreover, an increase in the mRNA level of *BnCRSH* was observed in roots; however, the differences were not significant (Figure 6D). For other variants, mostly a decrease in the transcription level of *BnRSHs* was observed. In the cotyledons of the seedlings that grew from the seeds inoculated with TvIV, TvVI, and TvVII, 3 times lower *BnRSH1* mRNA levels in comparison to the control were observed. In roots, the expression of *BnRSH1* was significantly down-regulated by inoculation with TvIII, TvIV, TvV, and TvVII, mostly by TvV, which reduced the expression 8 times in comparison to control (Figure 6A). The relative level of *BnRSH2* mRNA was significantly lower only in the cotyledons of the plants that grew from the seeds inoculated with TvVI, i.e., 2.3-fold lower than in the control. Significantly lower levels of *BnRSH2* were observed in the roots of the plants that grew from the seeds inoculated with TvIII, TvIV, TvVI, and TvVII, being approximately 3 times lower than in the control (Figure 6B). The level of *BnRSH3* in cotyledons was down-regulated in the seedlings that grew from the inoculated seeds; however, the differences were not significant (Figure 6C). In cotyledons, the expression of *BnCRSH* was reduced by TvIII, TvIV, TvVI, and TvVII; i.e., the expression was approximately 2 times lower than in the control seedlings (Figure 6D).

### 2.5. Field Experiment

The results of the field experiment showed that the inoculation of *B. napus* seeds with the spores of *T. viride* strain TvVII significantly affected the obtained yield, increasing it by 16.7% (Figure 7). *T. viride* strain TvVII also significantly increased the number of branches and siliques. There were on average two more branches per inoculated plant compared to the control, and the number of siliques was increased by 17.56% (Figure 7). There was an increase in the weight of thousand seeds (TSW) produced by the plants that grew from the inoculated seeds. Moreover, the inoculated plants were higher (5.75% higher) than the control plants; however, these differences were not statistically significant. 

## 3. Discussion

Progressive environmental pollution has a significant negative impact on living organisms, including the growth and development of crops. There is an urgent need to reduce the amounts of agrochemicals, and, thus, the development of effective biological solutions for agriculture is of crucial importance. The use of microbes in agriculture is an alternative approach to replace plant protection chemicals and fertilizers. Despite all belonging to the same species, the *T. viride* strains used in this study showed significant differences in mycelial morphology and the abilities to synthesize IAA and produce hydrolytic enzymes. The differences suggest that these isolates might also differ in the level of synthesis of other metabolites, such as elicitors, i.e., molecules that trigger plant immune responses [67]. The observed differences among the *T. viride* isolates probably explain the differences in the ability of the tested isolates to promote the growth of *B. napus* and the impact on *BnMT* and *BnRSH* gene expressions. It is widely observed that the positive impact of a single strain of microorganisms on plant growth and development depends on plant species or even variety. Moreover, several different strains of the same microorganism might affect the growth and development of the same plant differently, as observed in this study (reviewed in [68]). Interesting results were obtained by Zhang et al. [69]; i.e., three strains of the fungus *Mortierella elongata* varied in their effects on the height, dry biomass, and leaf area of watermelon (*Citrullus lanatus* Thunb. Matsum & Nakai), maize, tomato (*Solanum lycopersicum* L.), and pumpkin (*Cucurbita* sp. L.). In the present study, the inoculation of *B. napus* seeds with *T. viride* TvI–TvII and TvV–VII strains under laboratory conditions resulted in a significant increase in the length of the hypocotyls and roots of the seedlings. This effect might be due to the ability of those isolates to synthesize IAA; however, it needs to be noted that strain TvVII produced the lowest amount of IAA among the tested strains. As demonstrated by Contreras-Cornejo et al. [32], the mutation of *A. thaliana AUX1*, *BIG*, *EIR1*, and *AXR1* genes, which are involved in the transport and signaling of auxins, attenuated the growth-promoting effect of *Trichoderma virens* observed in wild-type plants. 

During plant–microorganism interactions, transcriptome changes occur both in the plant and in the microbe. We hypothesized that MTs and RSHs, which are widely known for their roles in the response to diversified environmental factors, are also important during plant interactions with microorganisms. The presence of many *cis*-regulatory elements regulating the response to abiotic and biotic stresses within the promoters of *MT* and *RSH* genes indicates that they participate in plant responses not only to environmental physio-chemical conditions but also to the presence of bacteria and fungi [40,42,44,56]. For example, the *MT* genes of rice and *A. thaliana* have numerous regulatory motifs of response to metals, low/high light, low temperature, and drought, as well as motifs associated with stress-related signaling, including the response to ROS and ABA [40]. In the case of *BnRSH*, the largest number of motifs was found to be associated with the response to abiotic stress and with signaling via phytohormones, including ethylene and ABA [56]. The analysis of *BnRSH* and *BnMT* promoter regions showed the presence of *cis*-regulatory elements responsive to microbial elicitors and other compounds of microbial origin. Thus, we assumed that the expressions of these genes might be altered in the presence of potential microbial symbionts, as well as in the presence of pathogens. Plant cells are equipped with pattern-recognizing receptors (PRRs), i.e., transmembrane proteins with ectodomains binding ligands of microbial origin—pathogen-/microbe-associated molecular patterns (P/MAMPs) [70]. The recognition of a ligand leads to the activation of plant pattern-triggered immunity (PTI), which limits the development of microorganisms in plant tissues, e.g., through an increased synthesis of antimicrobial compounds [71]. To date, a number of MAMP molecules produced by the fungi of the *Trichoderma* genus have been identified [72,73], including hydrolytic enzymes that allow for the penetration of the outermost layers of root tissues, such as cellulases [74], xylanases [75], and polygalacturonases [76]. A microscopic analysis showed progressive root colonization by germinating *T. viride* spores, including spore germination toward the root, the formation of a net-like structure, and the entwinement of root hairs by the growing hyphae. This may be due to the recognition of signaling molecules and/or nutrients (e.g., sugars) exuded by the roots and perceived by the fungus. The observed adhesion of the mycelium to the root is probably mediated by small proteins—hydrophobins [77]. It was shown that the presence of hydrophobins facilitates the colonization of plant tissue by fungi and probably protects the sensitive hyphal tips against the activity of antifungal compounds [78]. Moreover, within *BnMT* and *BnRSH* promoters, there were also elements of response to JA and SA—phytohormones responsible for triggering plant immune system responses. These mechanisms might be activated by the presence of *Trichoderma* fungi [36,37,79]. The observed changes in the expressions of *BnMT* and *BnRSH* genes are potentially the result of the changes triggered by the presence of the fungus. The presence of fungus is detected by plants because fungus secretes elicitors and hydrolytic enzymes that violate the integrity of the root tissue. Furthermore, changes in the balance of rapeseed phytohormones might further affect the expression of the analyzed genes. These conclusions are supported by the fact that, through the production of elicitors, fungi affect the level of accumulation of plant intermediate metabolites and by the increased signaling of SA, JA, ABA, or nitric oxide (NO) [80]. Within the promoters of the *BnMT1–BnMT3*, *BnRSH1*, and *BnRSH3* genes, the presence of motifs binding WRKY transcription factors was observed. An increase in the expression of genes encoding WRKY was demonstrated in response to the colonization of maize roots by *Trichoderma atroviride* [81]. Moreover, changes in the expression profiles of genes involved in JA and SA signaling in inoculated plants compared to the control were shown.

Genes encoding *MT1–3* in plants are expressed in various tissues and organs during ontogenesis. In most analyzed up-to-date species, *MT4* is expressed only in developing and mature seeds [82,83,84,85], and, thus, *BnMT4* was not analyzed in this study. It has been shown that *MT* expression changes in response to a number of endogenous and exogenous factors, i.e., the presence of heavy metals [86], ROS [87], polyamines [88], phytohormones [89], the presence of microorganisms [14,90,91], salinity [92], drought stress [43,48], osmotic stress [93], light conditions [94], and wounds and pathogen infections [95,96]. The presence of *T. viride* increased the expression of *BnMT1*–*3* in the cotyledons and roots of 6-day-old seedlings, but the effect was dependent on the type of *MT* and fungal isolate. Mirzahossini et al. [91] showed that the inoculation of *Festuca arundinacea* Schreb. with the fungus *Epichloe coenophiala* significantly increased plant tolerance to nickel, increased dry biomass, decreased H_2_O_2_ production, and lowered Ni accumulation in plant tissues. Moreover, *E. coenophiala* down-regulated the expressions of plant *MT* and ABC (ATP-binding cassette) metal transporter genes. The inoculation of rapeseed with arbuscular mycorrhizal fungi (AMF) (*Acaulospora longula*, *Glomus geosporum*, *G. mosseae,* and *Scutellospora calospora*) in sterile soil resulted in an increased concentration of N, P, K, and S in shoots but, at the same time, negatively affected plant growth compared to non-inoculated plants. Interestingly, in an experiment conducted in unsterile soil, the concentrations of analyzed elements were lower in the inoculated plants than in the control. The presence and absence of native microorganisms in the soil differentiated the effect on the expression of *BnMT2* inoculated with AMF spores; i.e., the expression was increased in the variant with sterile soil and decreased in the variant with unsterile soil containing native microorganisms [14]. In a study on the phytoextraction of heavy metals by willow (*Salix viminalis* L.) inoculated with the ectomycorrhizal fungus *Hebeloma mesophaeum* and bacteria *Bacillus cereus*, the content of Cd and Zn in plant tissues increased. Moreover, changes in *SvMT1* expression were observed. *B. cereus* significantly increased the level of *SvMT1* transcripts, whereas *H. mesophaeum* slightly decreased the expression of this MT [90]. The inoculation of rapeseed with the IAA-producing ectomycorrhizal fungus *Laccaria laccata* increased the number of leaves and the biomass of inoculated plants. In addition, *L. laccata* almost doubled *BnMT1* mRNA levels and decreased *BnMT3* transcript levels. Similar results have been observed in plants inoculated with *Collybia tuberosa* and *Cyathus olla* which significantly increased the level of *BnMT2* expression [97].

The stringent response was first observed in bacteria in response to amino acid deficiency [98], and *RSH* genes have been identified in bacteria [99,100,101], animals [102], and plants [103,104,105,106]. The stringent response genes are closely associated with plant adaptation to stress factors, including oxidative stress [107], nitrogen deficiency [62,106,107], salinity [108], drought, wounds, the presence of heavy metals, and UV radiation [53]. This is the first report showing the expression profiles of *RSH* genes in a plant in response to colonization by filamentous fungi. The tested fungal isolates caused an increase or a decrease in the expression of *BnRSH* genes depending on the type of *RSH* gene and fungal isolate. Interesting results were obtained for the *BnCRSH* gene. Although the *BnCRSH* promoter contains the highest number of regulatory elements responsive to biotic factors (3), and JA and SA signaling (8) among the tested *BnRSH*s, the expression of this gene decreased in response to colonization by fungi in cotyledons. To date, it has been shown that plant *RSH* genes are altered in the presence of rhizosphere bacteria. Dąbrowska et al. [56] showed that the inoculation of rapeseed with *Serratia plymuthica*, *Serratia liquefaciens*, and *Massila timonae* under salt stress conditions variably affects *BnRSH* expression. *S. plymuthica* significantly increased the expression of all analyzed genes, i.e., *BnRSH1*–*3* and *BnCRSH*, in roots and cotyledons. *S. liquefaciens* significantly increased *BnRSH1* and *BnCRSH* mRNA levels in canola cotyledons and *BnRSH1*–*3* expression in its roots, whereas *M. timonae* increased only the amount of *BnRSH1* transcripts in the root. The inoculation of canola with *P. stutzeri* significantly increased the expressions of *BnRSH1* and *BnRSH3* in roots. Moreover, an increased tolerance of inoculated canola to salt stress compared to non-inoculated plants was observed [108]. 

A major limitation of using bioinoculants is the low positive effect on plant growth and development in field experiments, despite their positive effects under laboratory conditions. This is due to the low level of survival of the microorganisms because of the competition for niches with native microorganisms. Moreover, the growth of microorganisms and/or their ability to promote plant growth highly depends on environmental factors, such as temperature, rainfall, and soil type [8]. For these reasons, the effectiveness of laboratory-developed bioinoculants must be further evaluated under field conditions. The inoculation of canola seeds with *T. viride* isolate TvVII spores positively affected *B. napus* growth and yield when grown in the field. Seed inoculation with *T. viride* TvVII spores increased plant tolerance to unfavorable weather. The inoculation of tomato with *Trichoderma* spp. spores increased plant height and biomass, as well as Ca, Mg, P, and K content, in shoots and roots under laboratory conditions. A larger area and number of leaves and a higher chlorophyll content were also noted compared with control plants [109]. In melon (*Cucumis melo* L.) inoculated with *Trichoderma phayaoense*, accelerated plant development, increased biomass, and increased yield were observed in laboratory studies. Moreover, *T. phayaoense* exhibited the ability to biocontrol the phytopathogens *Stagonosporopsis cucurbitacearum* and *Fusarium equiseti* [24]. The inoculation of turmeric (*Curcuma longa* L.) with *T. harzianum* resulted in increased plant height and yield under laboratory conditions. An analysis of *T. harzianum* revealed the ability of the fungus to solubilize P and to produce IAA, HCN, and cellulases [23]. *Trichoderma* fungi can also accelerate flowering, as shown in a greenhouse experiment on freesia (*Freesia refracta* Jacq. Klatt) inoculated with *T. viride*, *T. harzianum*, and *Trichoderma hamatum*. An accelerated development of side shoots and an increased content of K, Fe, Mn, and Zn were also observed [25]. It should be noted that, during the field experiment described in this publication, unfavorable environmental conditions prevailed throughout the canola growing season, i.e., ground frost, a low level of precipitation, and high air temperatures, which increased the effects of drought. Therefore, the fungal strain tested in this study effectively promoted plant growth during drought, which is now one of the most common and severe adverse environmental conditions. Under greenhouse conditions, it was shown that coating seeds with *T. harzianum* spores increased drought stress tolerance in wheat (*Triticum aestivum* L.). Plants colonized by the fungus showed lower levels of proline, malondialdehyde (MDA), and H_2_O_2_, as well as increased levels of phenolic compounds [110]. Similar results were obtained under laboratory conditions for maize inoculated with *T*. *atroviride* ID20G grown under drought-stress conditions. Increased biomass, chlorophyll, and carotenoid content, as well as limited lipid peroxidation associated with drought stress, were observed. An increased activity of antioxidant enzymes and decreased H_2_O_2_ content were also noted in inoculated plants [111]. A greenhouse study showed the increased activity of antioxidant enzymes in sugar cane (*Saccharum* sp. L.) inoculated with *Trichoderma asperellum* under drought conditions. Moreover, *T. asperellum* also increased photosynthesis efficiency and tissue water content [112]. There is still a knowledge gap in this research area, mainly regarding the effect of *Trichoderma* inoculation in field studies. Environmental conditions have a multifaceted effect on the microbial inoculation of plants, and the involvement of stress factors can alter the effectiveness of inoculation [8], which is why there is a need for the field testing of PGPM.

## 4. Materials and Methods

### 4.1. Microbial and Plant Materials

Seven morphologically different *Trichoderma viride* strains were used in this study (numbers in GenBank NCBI: OL221590.1, OL221591.1, OL221592.1, OL221593.1, OL221594.1, OL221595. 1, OL221596.1, respectively, hereinafter referred to as TvI–TvVII). The fungi used for the research were cultured on the microbial media PDA for a morphological analysis (potato dextrose agar, LABM) and TSB (trypticase soy broth, Sigma Aldrich; Poznań, Poland) for the assessment of the ability to synthesize IAA at room temperature. Seven-day-old cultures of the fungi growing on the PDA medium were used to prepare spore suspensions for the inoculation of *B. napus* seeds. Spores were collected by adding 5 mL of sterile dH_2_O to Petri dishes containing 7-day-old fungal cultures. With the use of sterile cell spreaders, the spores were suspended and then filtered via Miracloth (Merck; Warsaw, Poland) in order to remove hyphae. The concentration of the spores was then analyzed with the use of a Thoma cell counting chamber.

Seeds of spring rape (*Brassica napus* L.), variety Lumen, were purchased from Rapool (Wągrowiec, Poland). Before use, the seeds were surface-sterilized, according to the method presented by Mierek-Adamska et al. [113], in a mixture of 96% ethanol and 30% H_2_O_2_ (1:1, *v*:*v*) for 5 min and then rinsed at least 10 times with sterile water. The sterile seeds were incubated for 10 min in *T. viride* TvI–TvVII spore suspensions (10^6^ spores/mL for an in vitro study and 10^9^ spores/mL for a field experiment) with shaking (150 rpm) at room temperature. The control seeds were shaken in sterile water. Twenty seeds were placed in Petri dishes (90 mm diameter) on sterile filter paper moistened with 4 mL of sterile water. The seeds were incubated for six days in the dark at 25 °C. The length of the roots and hypocotyls and the fresh and dry biomass were analyzed after six days of incubation. For a gene expression analysis, the roots and cotyledons were frozen in liquid nitrogen and stored at −80 °C. 

### 4.2. Characterization of T. viride Isolates

#### 4.2.1. Morphological Analysis

Mycelium discs, obtained from the 7-day-old *T. viride* culture grown on solid PDA, were placed 1 cm from the edge of a Petri dish (90 mm diameter) on a solid PDA medium and cultured for 7 days at 25 °C. The following features of the mycelia were analyzed: fungal pigmentation, the presence of rings, the shape of conidia, the agglomeration of conidiophores in the form of lumps, and the ability to produce volatile compounds (a coconut smell).

#### 4.2.2. Evaluation of Capacity to Synthesize Indole-3-Acetic Acid

To verify the ability to synthesize indole-3-acetic acid (IAA), the spectrophotometric method based on Gravel et al. [114] was used. Three mycelium discs (for each strain) obtained from the 7-day-old *T. viride* cultures were placed in a liquid TSB medium supplemented with L-tryptophan at a final concentration of 200 µg/mL or without L- tryptophan. The cultures were then shaken for 7 days (150 rpm) at 24 °C. The cultures were then filtered through a sterile gauze to remove mycelia. Next, 4 mL of Salkowski’s reagent (0.5 M FeCl_3_, distilled H_2_O, and 70% HClO_4_ at a ratio of 2:49:49) was added to 2 mL of the filtrate and incubated for 20 min at room temperature. Absorbance was measured at a wavelength of 535 nm (NanoDrop, Thermo Fisher Scientific, Waltham, MA, USA). IAA reacts with ferric chloride and perchloric acid to give a colored product; i.e., the amount of colored compound was directly proportional to the concentration of IAA in the tested sample.

#### 4.2.3. Assessment of Hydrolytic Activity

The hydrolytic activity of the *T. viride* strains was analyzed based on a measurement of the hydrolysis zone around the colonies and the colony diameter of the examined fungi on the 7th day of cultivation at 24 °C. Enzymatic activity (Wact) was calculated according to Hrynkiewicz et al. [115]: Wact = Sh2/(Sc·t)(1)
where Sh—hydrolysis zone diameter; Sc—colony diameter; and t—incubation time (96 h and 168 h).

The ability to synthesize cellulase was assessed using a medium described by Wood [116] enriched with CMC (carboxymethylcellulose). Pectin hydrolysis was carried out on a medium described by Strzelczyk and Szpotański [117]. The level of pectin hydrolysis was verified by adding to the culture plates a solution of 1% cetyl-trimethylammonium bromide (BDH), which precipitates undigested pectin. 

### 4.3. Microscopic Analysis

Three-day-old *B. napus* seedlings grown on a water agar medium (Phytagel, Sigma-Aldrich) were inoculated with the *T. viride* strain TvVII spore suspension (10^6^ spores/mL), and, after 24 h, 48 h, and 72 h of cultivation, roots were collected. Next, the samples were placed on glass slides and covered with distilled water for observation using an Olympus BX50 fluorescence microscope. The results were recorded with an Olympus XC50 digital color camera and Cell^B^ software, version 3.4 (Olympus Soft Imaging Solutions GmbH; Münster, Germany). For the control, the inoculation of plants with *T. viride* spores was omitted. 

### 4.4. Analysis of Gene Expression

Total RNA was isolated from 100 mg of the roots and cotyledons of 6-day-old canola seedlings according to Chomczyński and Sacchi [118]. The obtained RNA was analyzed via spectrophotometric measurements (NanoDrop Lite, Thermo Fisher Scientific; Waltham, MA, USA) and electrophoresis in agarose gel stained with ethidium bromide. High-quality RNA samples with an A_260_/A_280_ ratio of approximately 2.0 were used for further analysis. For the reverse transcription reaction, 1.5 μg of total RNA was used, and the reaction was performed according to Dąbrowska et al. [97].

The relative levels of *BnRSH* and *BnMT* gene transcripts were checked using the semiquantitative (sq) RT-PCR method [119]. The relative level of the transcripts of the tested genes is expressed as the ratio of the amount of the RT-PCR product for the analyzed gene to the amount of the RT-PCR product for the reference gene encoding actin (*BnAct*). The reaction mixture contained 1 μL cDNA, 2 μL Opti*Taq* polymerase buffer, 0.2 mM dNTP, 0.15 μM forward and reverse primers, and 1 U Opti*Taq* DNA polymerase (EURx, Gdańsk, Poland) in a final volume of 20 μL. The reaction conditions were as follows: initial denaturation at 95 °C for 2 min, denaturation at 95 °C for 30 s, primer annealing for 45 s (temperature optimized for each primer pair given in Table 4), elongation at 72 °C for 35 s, and final elongation at 72 °C for 5 min. The number of cycles was optimized for each pair of primers to complete the reaction in a logarithmic growth phase (Table 4). The sequences of the oligonucleotides used are listed in Table 1. ImageGuage 3.46 software (FujiFilm; Tokyo, Japan) was used for densitometric measurements of the amount of obtained RT-PCR products.

### 4.5. Field Experiment

The field experiment was conducted in Chrząstowo (53°09′52″ N 17°35′02″ E), on a site with typical brown soil, medium clay, and soil pH 6.4. The content of bioavailable minerals was as follows: Mg—47 mg/kg^−1^, P—95 mg/kg^−1^, and K—183 mg/kg^−1^ (Chemical-Agricultural District Station in Bydgoszcz, Poland). Canola seeds inoculated with *T. viride* strain TvVII and non-inoculated *B. napus* seeds were used for the field experiment. The fore crop was spring barley. The size of the experimental field was 16.5 m^2^, and the seeds were sown at a density of 100 plants per m^2^. The experiment was carried out in three replicates on different plots simultaneously. Fertilization was carried out at half-dose before sowing, and the other half was used afterward as a top dressing, with total doses of the pure ingredients of 30 NH_4_NO_3_ kg/ha; 48 P_2_O_5_ kg/ha; 34 (NH_4_)_2_SO_4_ kg/ha; and 80 K_2_O kg/ha. The insecticides Decis Mega 50EW (Bayer; Leverkusen, Germany) and Boravi 50 WG (Gowan; Yuma, AZ, USA) were used to protect the plants against silique pests, including *Meligethes aeneus*. 

The experiment was established on 7 April 2022, and the crop was harvested on 12 August 2022. During the growing season, there were unfavorable environmental conditions, i.e., ground frost in April (down to −4.5 °C) and May (down to −6.2 °C), and high air temperatures (up to 34.6 °C in June and up to 35.7 °C in July). The precipitation during the growing season is given in Table 5. The following plant parameters were evaluated: the number of branches and siliques, the height of the plants (10 randomly selected plants in three replicates), the seed yield determined at 9% humidity of seeds, and thousand seed weight (TSW). 

### 4.6. Statistical Analysis

Statistical analyses (one-way ANOVA with Tukey’s *post hoc* test) were performed in Past 4.12b [120]. For most analyses, the threshold for significance was *p* < 0.05. For the results of the field experiment, the thresholds for significance were *p* < 0.05 (*), *p* < 0.01 (**), and *p* < 0.001 (***). 

## 5. Conclusions

The development of novel bioinoculants for agriculture is crucial to reduce the usage of chemical plant protection products and mineral fertilizers while maintaining high yields and quality. However, advancing global warming and the more frequent occurrence of drought periods require intensified efforts for the betterment of crop systems and novel solutions in agriculture. These are the most important reasons for further and deeper research on interactions between plants and microorganisms, especially in economically important crops. In order to develop bioinoculants more effectively, the molecular basis, i.e., the proteins and pathways underlying plant–microbe interactions, needs to be thoroughly understood. The conducted research shows that the saprophytic fungus *T. viride* colonizes the roots of the non-mycorrhizal plant *B. napus*. Furthermore, it accelerates germination and growth, as well as increases the yield. In addition, as the field study showed, the presence of *T. viride* protects rapeseed plants against water deficits. Molecular studies showed changes in the expressions of *B. napus MT* and *RSH* genes in response to the presence of *T. viride*; however, the changes depended on the tested fungal strain. This indicates the involvement of metallothioneins and the stringent response proteins in plant–fungus interactions. The results obtained in this study will contribute to the development of novel bioinoculants (*T. viride* VII) for the direct or indirect regulation of plant metabolism, crucial for food security. The conducted research shows that each fungal strain has a different potential to interact with the selected plant. The information obtained in this study is particularly important for the producers of biopreparations, as it raises the awareness that the microorganisms that are biocomponents of effective biopreparations should be thoroughly tested.

## Figures and Tables

**Figure 1 ijms-24-15349-f001:**
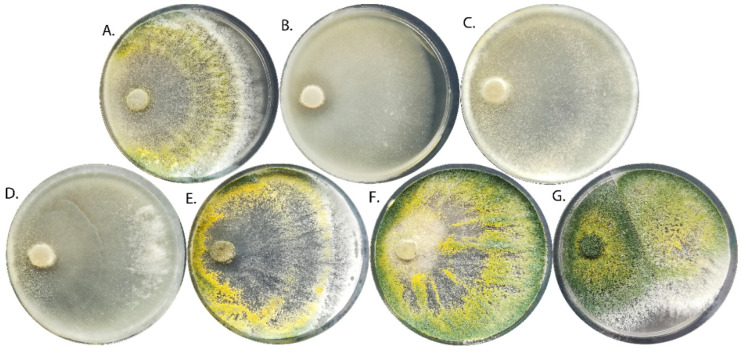
Morphology of 7-day-old mycelium of different strains of *T. viride* ((**A**)—TvI, (**B**)—TvII, (**C**)—TvIII, (**D**)—TvIV, (**E**)—TvV, (**F**)—TvVI, (**G**)—TvVII) on PDA medium.

**Figure 2 ijms-24-15349-f002:**
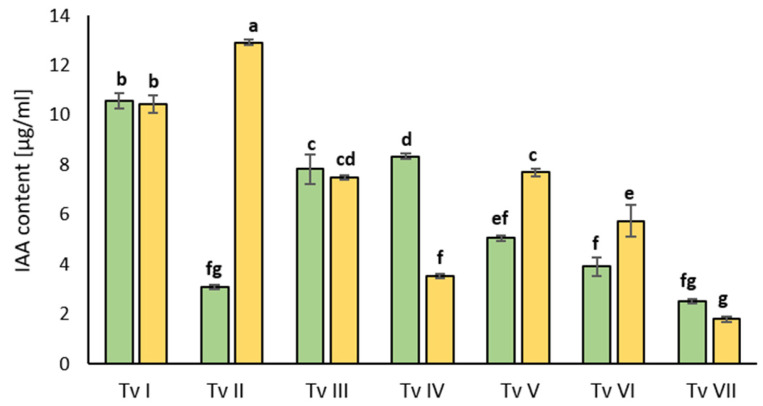
Levels of IAA produced in liquid cultures of different strains of *T. viride* (TvI-TvVII) not supplemented with L-tryptophan (green bars) and supplemented with L-tryptophan (yellow bars). Bars represent the mean of three replicates ± SD. Different letters indicate statistically significant differences between the tested variants at *p* < 0.05 (one-way ANOVA with *post hoc* Tukey’s test).

**Figure 3 ijms-24-15349-f003:**
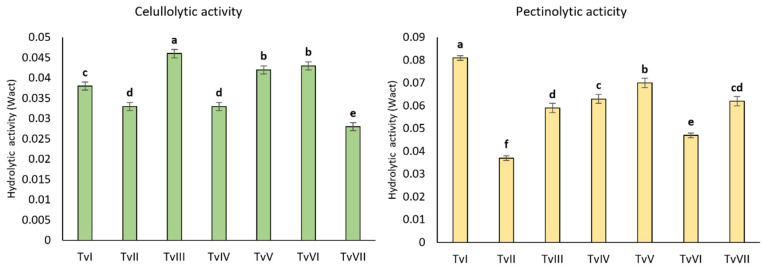
The hydrolytic activity of different strains (TvI-TvVII) of *T. viride*. Bars represent the mean of three replicates ± SD. Different letters indicate statistically significant differences at *p* < 0.05 (one-way ANOVA with *post hoc* Tukey’s test).

**Figure 4 ijms-24-15349-f004:**
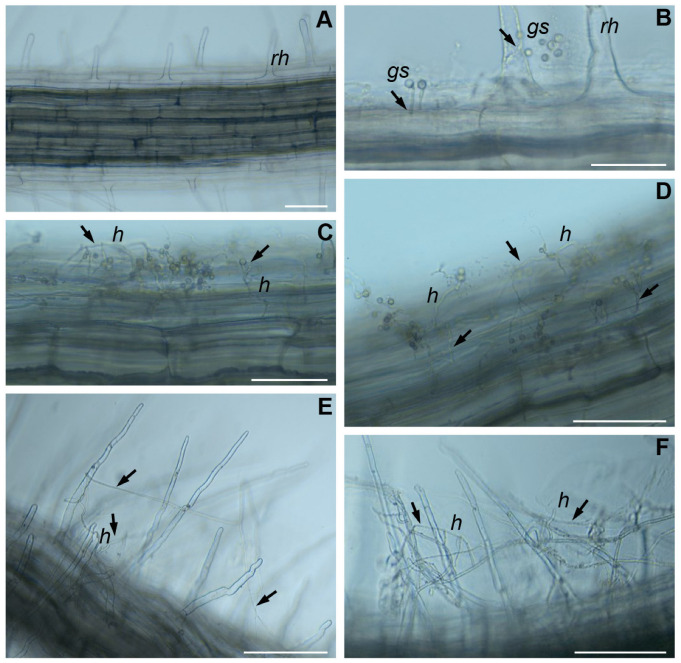
Progressive colonization of *B. napus* seedling root by *T. viride* strain TvVII. (**A**) Root of control non-inoculated seedling; (**B**) 24 h after inoculation, germinating spores (arrows) of *T. viride* on root and root hairs surface were observed; (**C**,**D**) intensive *T. viride* mycelial growth and root penetration (arrows) 48 h after inoculation; (**E**,**F**) 72 h after inoculation, coiling of *T. viride* hyphae (arrows) around root hairs was visible. rh—root hair, gs—germinating spores, h—hyphae; scale bars represent 100 µm.

**Figure 5 ijms-24-15349-f005:**
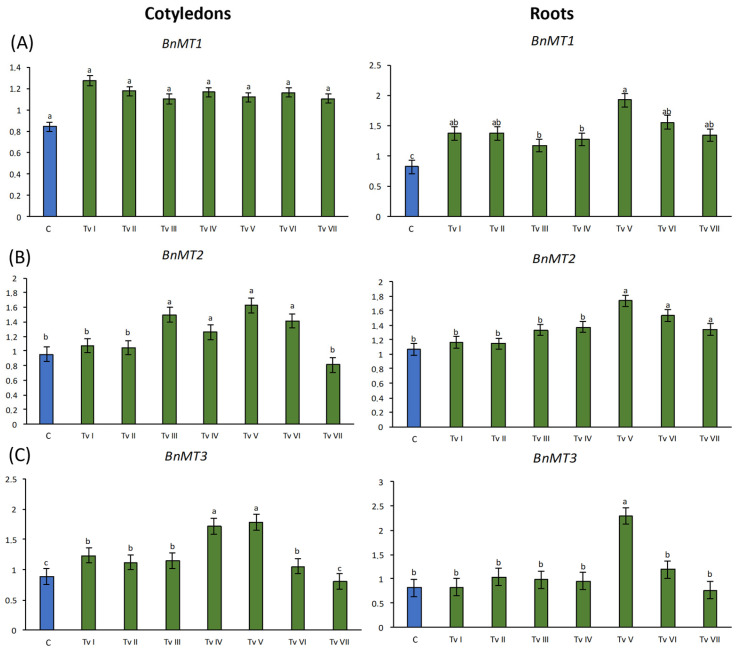
Relative levels of *BnMT1* (**A**), *BnMT2* (**B**), and *BnMT3* (**C**) transcripts in cotyledons and roots of 6-day-old *B. napus* seedlings that grew from seeds inoculated with *T. viride* spores of 7 strains (TvI–TvVII) and control (**C**). Bars represent mean values ± SD. The *y*-axis represents the relative transcript level expressed as the ratio of the amount of RT-PCR product of the analyzed gene to the amount of RT-PCR product of the reference gene. Different letters indicate statistically significant differences at *p* < 0.05 (one-way ANOVA with *post hoc* Tukey’s test).

**Figure 6 ijms-24-15349-f006:**
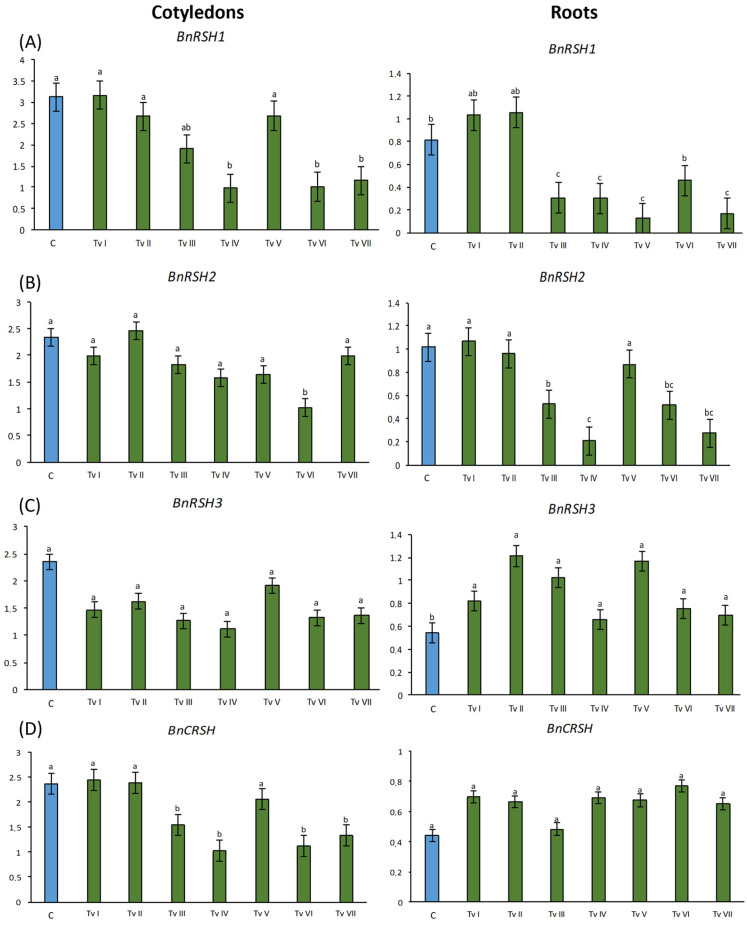
Relative mRNA levels of *BnRSH1* (**A**), *BnRSH2* (**B**), *BnRSH3* (**C**), and *BnCRSH* (**D**) genes in cotyledons and roots of 6-day-old *B. napus* seedlings that grew from seeds inoculated with *T. viride* of seven strains (TvI–TvVII) and control (**C**). Bars represent mean values ± SD. The *y*-axis represents the relative transcript level expressed as the ratio of the amount of RT-PCR product of the analyzed gene to the amount of RT-PCR product of the reference gene. Different letters indicate statistically significant differences at *p* < 0.05 (one-way ANOVA with *post hoc* Tukey’s test).

**Figure 7 ijms-24-15349-f007:**
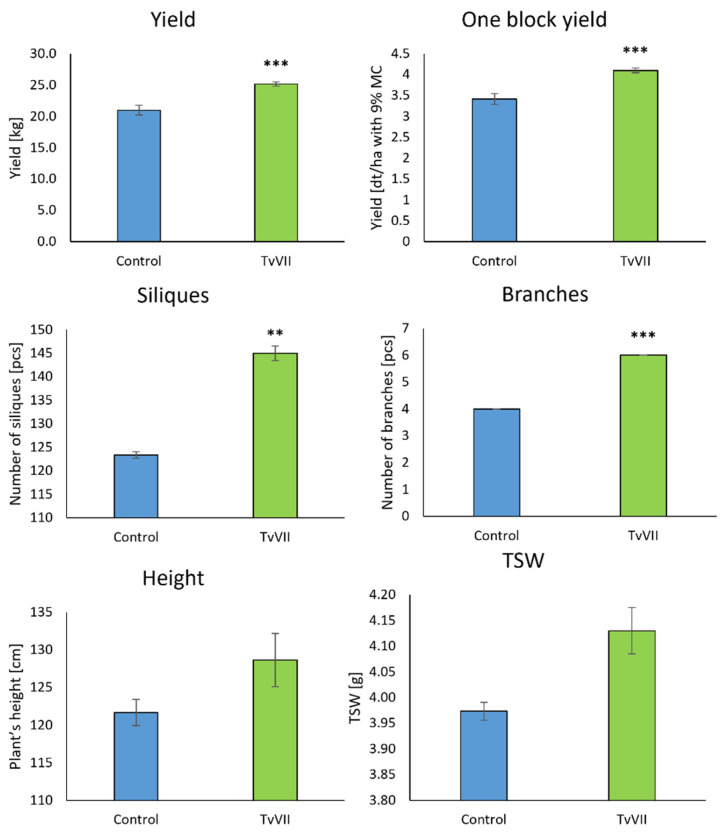
Evaluation of biometric parameters of *B. napus* plants that grew from seeds inoculated with *T. viride* VII (green) and control plants (blue) grown under field conditions. Asterisks indicate significant differences at *p* < 0.01 (**) and *p* < 0.001 (***) (one-way ANOVA with *post hoc* Tukey’s test).

**Table 1 ijms-24-15349-t001:** Morphological characteristics of the mycelia of different strains (TvI-TvVII) *T. viride* grown on PDA medium.

Strain	Pigmentation of Mycelium	Concentric Rings	Coconut Smell	Conidiophore Agglomeration	Shape of Conidia
TvI	Yellow-green-white	+	-	+	Spherical
TvII	White	-	+	-	Elliptical
TvIII	White	-	+	-	Spherical
TvIV	White	-	-	-	Spherical
TvV	Yellow-green-white	+	+	+	Spherical
TvVI	Yellow-green-white	+	-	+	Spherical
TvVII	Dark green	-	-	+	Spherical

**Table 2 ijms-24-15349-t002:** Effect of inoculation of *B. napus* seeds with *T. viride* spores of different strains (TvI–TvVII) on the growth and development of 6-day-old rapeseed seedlings. Non-inoculated 6-day-old seedlings served as a control (C). Numbers represent mean values (*n* = 90) ± SD. Different letters indicate statistically significant differences at *p* < 0.05 (one-way ANOVA with *post hoc* Tukey’s test).

Strain	Root [cm]	Hypocotyl [cm]	Fresh Biomass [mg]	Dry Biomass [mg]
C	1.8 ± 0.2 b	2.6 ± 0.2 a	39.22 ± 1.48 c	3.84 ± 0.63 b
TvI	2.5 ± 0.2 a	3.2 ± 0.2 a	36.78 ± 1.25 c	3.55 ± 0.09 b
TvII	2.4 ± 0.2 a	3.9 ± 0.2 a	41.52 ± 1.34 c	3.70 ± 0.13 b
TvIII	1.9 ± 0.1 b	1.7 ± 0.1 b	34.06 ± 0.81 c	4.03 ± 0.09 b
TvIV	2.2 ± 0.1 b	1.6 ± 0.1 b	32.53 ± 1.12 c	3.63 ± 0.12 b
TvV	2.2 ± 0.2 b	3.4 ± 0.2 a	43.06 ± 1.47 c	3.80 ± 0.17 b
TvVI	2.3 ± 0.2 a	3.1 ± 0.2 a	50.72 ± 1.86 a	4.53 ± 0.86 a
TvVII	2.6 ± 0.2 a	3.5 ± 0.3 a	43.58 ± 1.09 b	4.63 ± 0.10 a

**Table 3 ijms-24-15349-t003:** *Cis*-regulatory elements involved in the response to biotic factors and phytohormones (JA and SA) signaling within the promoters of the analyzed *BnMT* and *BnRSH* genes.

	Motif	Genes and the Number of Elements	Function
** *BnMT* **	TC-rich repeats	*BnMT1* (1), *BnMT3* (1)	Defense and stress responsiveness
	W box (TTGACC)	*BnMT1* (1), *BnMT2* (1), *BnMT3* (2)	WRKY binding site involved in wounding and pathogen response
	CGTCA-motif	*BnMT2* (1), *BnMT3* (4)	MeJA responsiveness
	TGACG-motif	*BnMT2* (1), *BnMT3* (4)	
	TCA-element (CCATCTTTTT)	*BnMT1* (1), *BnMT2* (1)	SA responsiveness
	TCA (TCATCTTCAT)	*BnMT3* (3)	
	as-1 (TGACG)	*BnMT2* (1), *BnMT3* (4)	
** *BnRSH* **	AT-rich sequence	*BnCRSH* (1)	Fungal elicitor-mediated activation
	TC-rich repeats	*BnCRSH* (2)	Defense and stress responsiveness
	W box (TTGACC)	*BnRSH1* (1), *BnRSH3* (1)	WRKY binding site involved in wounding and pathogens response
	CGTCA-motif	*BnRSH2* (2), *BnRSH3* (3), *BnCRSH* (4)	MeJA responsiveness
	TGACG-motif	*BnRSH2* (2), *BnRSH3* (3), *BnCRSH* (4)	

**Table 4 ijms-24-15349-t004:** Oligonucleotides used in sqRT-PCR and sqRT-PCR reaction conditions.

Gene/Gene ID	Primer Sequences 5′-3′	Size of PCR Product (bp)	Temperature of Annealing	Number of PCR Cycles
** *BnMT1* ** **JX035784.1**	for TGGCAGGTTCTAACTGTGGA rev CAAATGAAAACATTATACACCACACA	309	52 °C	30
** *BnMT2* ** **JX103200.1**	for TCAATTTGATTAAATTCTCTGCT rev AAGCCTGCAGCCATTATTACA	401	52 °C	30
** *BnMT3* ** **JX103201.1**	for GCAAAACAACAAAACACACACA rev CTCACGCTATCCTCCGTCTC	418	56 °C	26
** *BnRSH1* ** **XM_013821537.3**	for GGAGGTTCAGATCAGAACCG rev CCATTCACCTTCGCTGCTAC	396	58 °C	32
** *BnRSH2* ** **XM_022703426.2**	for GCAAGATGTTGAAGAAGAATCTAACG rev GCACAGACATCTTGTCATTTTCG	534	54 °C	34
** *BnRSH3* ** **XM_048781742.1**	for CCGAAACTTTCCGATTTCAA rev TCGTAGTCAACGCACGAGTC	524	54 °C	34
** *BnCRSH* ** **XM_048753141.1**	for AAGTGATGGAGGAGCTTGGA rev CCATTTACTGGAACGCAACA	263	54 °C	38
** *BnAct* ** **NM_001316010.1**	for CTCACGCTATCCTCCGTCTC rev TTGATCTTCATGCTGCTTGG	469	52 °C	30

**Table 5 ijms-24-15349-t005:** Precipitation during rapeseed growing season.

Month	I Decade [mm]	II Decade [mm]	III Decade [mm]	Sum [mm]
April	10.6	13.2	0.7	24.5
May	2.2	6.7	16.5	25.4
June	8.6	23.5	7.4	39.5
July	27.6	9.2	6.1	42.9

## Data Availability

Relevant data applicable to this research are within the paper and are available on request from the corresponding author.

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
