# Peer review of "Trichoderma viride Colonizes the Roots of Brassica napus L., Alters the Expression of Stress-Responsive Genes, and Increases the Yield of Canola under Field Conditions during Drought"

_ijms, 2023, doi:10.3390/ijms242015349_

Round 1

Reviewer 1 Report

I have an opportunity to review manuscript : „Trichoderma viride colonizes the roots of Brassica napus L., alters expression of stress responsive genes, and increases the yield of canola in field conditions during drought” submitted to IJMS.

Authors concentrated on promoting the growth of canola plants and potentially presented Trichoderma viride on growth condition and root system formation. Some parts concentrated on genes expression showed quite interesting and promising data, but many aspects should be explained and clarified, I presenting the list of the main issues below:

Honestly, presenting results are predictable and the main statement that Trichoderma positively regulates and influenced on Brassicaceae growth and other plants families are commonly known and already published scientific fact. Moreover, Trichoderma treatment in agriculture and horticulture makes plants more resistance to e.g. fungal pathogen and more resistance to abiotic drought or heat stress. Therefore, one of the problem with submitted manuscript is low novelty aspects and predictable results.

Much more interesting will be transcriptomic analyses revealed which components will be induced and which down-regulated in these experimental condition, than proposed selected genes expression. The question is and it should be more deeply explained - why Authors decided to analyse metallothionein’s genes in the context drought stress, the reader should have these information before he or she will be analyse the results ?? (I know that using PlantCare system gives some path).

On the other side selection of RSH genes was clearly explain and these part of the results is quite interesting.

The introduction part give the reader sufficient background to analyze the obtained results;

Figure 2 and figure4 and 6- please, add using statistical test in figures caption;

Table 2- I suggest to reorganized these analyses to presenting by chart- It will be much more clear for the reader;

I do not agree with the statement “Physiological parameters of seedling” section 2.2 – Presenting findings are only plant growth versus biomass parameters- to be honest, physiological parameters were a more interesting take of the subject here;

Figure 3- these analyses revealed very poor in quality of obtained microscopical documentation- photo A, C and D had the worse quality, that’s like 'overexposed'– For example, where is “cell wall damage in the presence of TvVII as indicated by arrows 24 h after inoculation”?? Please, improve the resolution and quality of photographic documentation. There are something wrong with scale bars proportion between different resolution (Sorry, but presenting scale bars look like copy-paste in hurry);

What kind of reference genes were used to qPCR analysis? The transcripts levels were relative to what kind of two reference genes? – I see only one actin - what I guess by table presenting primers - it can be not enough for testing different stress conditions, because it is known facts that it is not the most stable (despite of being used in literature).

Please rethink, if wouldn’t be more clear to present for roots and cotyledons two charts for methalothioneins and the same situation for RSH genes – it is possible that when to the charts will be added letters  (A,B, C …) it will be easier to cite them in the text-  not used the “second graph form the top" and so on);

Part 2.5  - agronomic parameters – this is predictable effect, treatment seeds with different Trichoderma species improved plant yield and quality, especially TSW parameters;

English language required extensive revision;

Author Response

Dear Reviewer,

Kind regards

Reviewer 2 Report

The manuscript entitled “Trichoderma viride colonizes the roots of Brassica napus L., alters expression of stress responsive genes, and increases the yield of canola in field conditions during drought” is well crafted by the authors. I appreciate authors extensive work, presentation, and research contribution to an important field of agriculture (stress biology).

However, I wish to highlight few points which must be addressed for further perusal-

     There are no comments with Introduction, and discussion , they are sufficiently elaborated - the hypothesis is clearly defined-

Materials and methods

1.      Authors have described about the fungal growth on PDA and other media but there were no mentions of  what temperature, and if used incubator or just grew at room temperature, all minute details are necessary to be included—secondly there was mention of spore suspension preparation, how it was prepared. Please elaborate all along with citation if needed.

2.      Why 96% ethanol was used for surface sterilization, and 30 % H2O2? There was also no mention of time of exposure, please justify with citation, it might be toxic to seeds - please justify.

3.      Field experiment- if authors have data for more soil characteristics such as total N,P,K availability, organic matters etc. will be great to add on.  

4.      The crop was harvested but there was no inclusion of what data was collected ? Please include it in the methods section.

Results-

5.      The figures of field experiment have no statistical letters , please work on it as it will be better to understand the differences—

6.      There was also no mention if authors have checked colonization of TV in field samples. The crops completed its full tenure of growth, but what made authors sure that in field conditions TV only worked, as there is other native microbial flora too, TV has to compete to survive. Also do authors have managed to access the survivability of TV till the end of experiment by soil or rhizosphere/ root analysis ? please justify –

7.      There are also no conclusions for the article, it is better if authors include it.

Author Response

Dear Reviewer,

Regards

Round 2

Reviewer 1 Report

Dear Editor and Authors,

-Some suggestions were corrected by Authors like table 2 converted to chart, adding statistical tests to the figure captions, statement “physiological parameters” was converted to the statement “growth parameters”.

Firstly, It was a very good decision to add more information about metallothionein’s role in a wide plant- stress condition environment, because in previous version these role was as veiled as it was possible. Unfortunately, Authors still do not explained their role/ involvement in Trichoderma-drought stress condition, despite of fact that they aimed explaining the potential role of MTs and RSHs in plant-microbe interaction – we have only confirmation of relative expression changes in several BnMT1,2,3; I’m of the opinion that analyses conducted based on knockout and/or overexpressed mutants can give us the real explanation;

Unfortunately, I do not agree with Authors statement that :” , this is the first report on the use of Trichoderma viride for canola during drought”- because when we analyzed the current literature only form last few years many aspects are commonly known – I can recommend some recent papers from the wide context of Brassica or different Brassicaceae reaction in multistress conditions -Trichoderma and drought stress

For example: Poveda et al., 2020; Mohan et al., 2021; also on canola seeds- Cardarelli et al., 2022; Purwanto et al., 2022 as well as  very interesting review papers summarizing the recent knowledge with wide aspects of Trichoderma effect in drought stress presented by Boorboori and Zhang in 2023;

Additional comments – As a plant geneticists with experience the Authors probably know that choosing only one reference genes can be insufficient for describing precise relative genes expression;

Moreover, current Figure 4 – scale bars were corrected, but photos are still overexposed and with low quality;

I do not change my opinion about low novelty aspects and predictable especially growth-condition results,

Sincerely

Some minor English corrections are nedded;

Author Response

Dear Reviewer,

Thank you for your review. We are sending answers to your comments and questions.

-Some suggestions were corrected by Authors like table 2 converted to chart, adding statistical tests to the figure captions, statement “physiological parameters” was converted to the statement “growth parameters”.

  1. Reviewer: “Firstly, It was a very good decision to add more information about metallothionein’s role in a wide plant- stress condition environment, because in previous version these role was as veiled as it was possible. Unfortunately, Authors still do not explained their role/ involvement in Trichoderma-drought stress condition, despite of fact that they aimed explaining the potential role of MTs and RSHs in plant-microbe interaction – we have only confirmation of relative expression changes in several BnMT1,2,3; I’m of the opinion that analyses conducted based on knockout and/or overexpressed mutants can give us the real explanation;”

Answer: We agree with the statement made by the Reviewer. Better results would be obtained from research on the mutant plants. However, in the case of canola, the generation of knock-out mutants is not as simple as in the case of model plants such as Arabidopsis thaliana. We have already undertaken research to develop an efficient way to generate knock-out canola plants, and the members of our team are currently conducting time-consuming experiments. The preliminary results have been presented at scientific conferences. (Boniecka J., Turkan S., Nekrasov V., Ruttnik T, Eeckhaut T., van Laere K. Targeted mutagenesis in oilseed rape (Brassica napus L.) protoplasts using CRISPR/Cas. Plant genome editing: the wide range of applications: 2nd PlantEd Conference, 20-22 September 2021, Lecce, Italy, s. 54; Boniecka J., Rybicka A., Sweigchofer A., Trejgell A. Rape (Brassica napus L.) transformation and shoot organogenesis : important steps towards successful genome editing Plant genome editing state of the art: 1st PlantEd Conference 2019, Novi Sad).

In the presented manuscript, we only point out the direction of the research on the role of the stringent response and genes involved in plant metal homeostasis—metallothioneins, whose role in the response of plants to drought stress is undeniable. We are aware that we are at the stage of preliminary studies, but still, our results, including detailed microscopic analysis of the establishment of the interaction between the non-mycorrhizal crop plant and the filamentous fungus, are novel. So far, we have not found any publication showing the colonization of spring canola by the Trichoderma viride. Similarly, there is no publication that demonstrates the involvement of the stringent response genes encoding important stress proteins and metallothioneins, which have documented antioxidative role, in the canola-T. viride interactions. Important information from the analysis of the gene expression is the fact that each of the T. viride isolates exerts a different effect not only on the mRNA level but also on the growth parameters. It is difficult to define the role of these genes in plant-microbe interactions just through gene expression analysis, as this would require a comprehensive study on protein level in plant-microbe interactions.

We would like to express that particularly important to us and other researchers in the presented manuscript are the field study results, which show the actual impact of T. viride on the growth and yield of spring canola, which may have a real impact on practical aspects of cultivating canola.

  1. Reviewer: “Unfortunately, I do not agree with Authors statement that :” , this is the first report on the use of Trichoderma viride for canola during drought”- because when we analyzed the current literature only form last few years many aspects are commonly known – I can recommend some recent papers from the wide context of Brassica or different Brassicaceae reaction in multistress conditions -Trichoderma and drought stress – 

For example: Poveda et al., 2020; Mohan et al., 2021; also on canola seeds- Cardarelli et al., 2022; Purwanto et al., 2022 as well as  very interesting review papers summarizing the recent knowledge with wide aspects of Trichoderma effect in drought stress presented by Boorboori and Zhang in 2023;”

Answer: We thank the Reviewer for pointing out new important and interesting articles in the field described in the presented manuscript. We are aware that this particular research field on different Trichoderma fungi on an important group of crop plants - Brassica, was and is extensively studied. We are also aware of the fact that each isolate of a particular Trichoderma species can have a very different impact, or no impact at all, on canola or other Brassica plants. Furthermore, differences in the exerted effects on particular plant species are more pronounced in the case of the application of different species of Trichoderma. These, important information justifies conducting continuous research on Trichoderma-Brassica, especially if we consider other factors that affect these interactions, i.e., different soil types, different soil microbiomes, climate, the presence or lack of stress factors, and different cultivars of the studied plant. This approach guarantees novelty in this field. After careful consideration of the suggested by the Reviewer articles, we found that, these information:

  • Poveda et al., (2020) concern the interaction of napus cv. Jura with different species of Trichoderma (T. parareesei), and the study was conducted as a pot experiment.
  • Mohan et al., (2021) concern the interaction of different species of Brassica ( juncea) with Trichoderma viride and Bacillus subtilis. The experiment was conducted in the field, and the authors used soil application method.
  • Other proposed articles are reviews.
  1. Reviewer: Additional comments – As a plant geneticists with experience the Authors probably know that choosing only one reference genes can be insufficient for describing precise relative genes expression; 

Answer: Naturally, we agree with this comment, but at the moment we would like to point out that this is very common practice, observed in many published articles on gene expression analysis – based on the single reference gene. Here are few examples: Poveda, 2020 (https://doi.org/10.3390/agronomy10010118), Bailey et al., 2006 (https://doi.org/10.1007/s00425-006-0314-0), Zhane et al., 2016 (https://doi.org/10.3389/fpls.2016.01405), Alkooranee et al., 2015 (https://doi.org/10.1371/journal.pone.0142177). Zhang et al., 2019 (https://doi.org/10.1186/s12870-018-1618-5).

  1. Reviewer: Moreover, current Figure 4 – scale bars were corrected, but photos are still overexposed and with low quality;

Answer: We have repeated the experiment and took a series of pictures showing canola-T.viride interaction.

  1. Reviewer: I do not change my opinion about low novelty aspects and predictable especially growth-condition results,

Answer: We understand that it is difficult to change the first impression; however, based on our explanations addressing the concerns, we believe that it is possible after a thorough re-reading of the manuscript, with particular attention paid to the relatively rare field study results.

Kind regards,

Authors

Round 3

Reviewer 1 Report

Dear Authors,

First of all, as I see during the revision process one new Co-Author was added;

The visualization of Brassica napus seedling roots colonized by T. viride was significantly improved;

English language correction was done;

I received „improved” version of the manuscript with working-comments, taking into account the authors' country of origin, I guess in Polish language?

Unfortunately, I cannot agree with the opinion and some Author’s explanations:

-it is not an scientific explanation: it is common practice, therefore we used one reference gene;

-As I had said before: I do not agree with Authors statement that :” , this is the first report on the use of Trichoderma viride for canola during drought, but I will be honest, I did not expect so long and winding answer, but simply tone down this statement;

-It is really very nice to hear that Authors work on further interesting studies, but due to reviewer’s obligation I evaluate these particularly obtained results;

To sum up- I am not in the habit of blocking manuscript from publication, but the authors' statements are insufficient.

Sincerely

Minor editing of English language required.